# Whole Slide Image Domain Adaptation Tailored with Fisher Vector, Self Supervised Learning, and Novel Loss Function

**Ravi Kant Gupta**[*] ID                                                  RAVIGUPTA131@GMAIL.COM
**Shounak Das**[*] ID                                                      21D070068@IITB.AC.IN
**Shambhavi Shanker**[*]                                                   21D070066@IITB.AC.IN
**Ardhendu Sekhar**                                                        214070020@IITB.AC.IN
**Amit Sethi**                                                             ASETHI@IITB.AC.IN

*MeDAL Lab, Department of Electrical Engineering, IIT Bombay, India*

## Abstract

Whole Slide Images (WSIs) present major challenges in computational pathology due to their high resolution, morphological diversity, and domain variability across institutions. These factors result in domain shifts that limit model generalization. We propose a domain adaptation framework that integrates self-supervised learning, clustering, and Fisher Vector encoding for robust WSI classification. Patch-level features are extracted using MoCoV3, clustered via k-means, and aggregated using Gaussian mixture-based Fisher Vectors to form compact slide-level representations. To align domains, we employ adversarial training enhanced with a tailored loss combining PLMMD and MCC. Evaluated on HER2 classification across TCGA-BRCA and Warwick datasets, our method consistently outperforms baselines, especially under label noise and domain shift, demonstrating the strength of combining self-supervised features with structured statistical encoding for cross-domain WSI analysis.

**Keywords:** Domain Adaptation, Fisher Vector, Self-supervised, Whole Slide Image.

## 1. Introduction

Whole Slide Images (WSIs) offer gigapixel-resolution views essential for clinical diagnosis (Komura and Ishikawa, 2018), but their size, morphological diversity, and inter-institutional variability pose significant computational and generalization challenges. WSIs are typically divided into patches for processing, which disrupts spatial context. Domain shifts arising from differences in staining and imaging protocols (Ganin and Lempitsky, 2015), coupled with limited annotations, call for weakly supervised, domain-adaptive models.

We propose a framework that combines self-supervised learning (SSL) with Fisher Vector (FV) encoding for robust slide-level representation. Patch features are extracted using MoCo, clustered via k-means, and encoded using GMM-based FVs. To address domain shift, we apply adversarial training enhanced with PLMMD (Gupta et al., 2025) and MCC (Abhishek and Hamarneh, 2021) losses. Evaluated on HER2 classification with TCGA-BRCA (Genomic Data Commons, 2024) and Warwick (Qaiser et al., 2018), our method achieves strong cross-domain performance. The approach is modular and label-efficient, making it well-suited for real-world WSI analysis.

---

[*] Contributed equally

Gupta Das[*] Shanker[*] Sekhar Sethi

## 2. Methodology

We address WSI classification under domain shift by combining self-supervised feature extraction, clustering, Fisher Vector encoding, and adversarial adaptation. Quality control removes artifacts from TCGA-BRCA and Warwick datasets using methods from (Patil et al., 2023; Gupta et al., 2023), and each WSI is tiled into $512 \times 512$ patches at 40x magnification. Patch embeddings are extracted using MoCoV3 (Chen et al., 2021), which generalizes better across domains than supervised ResNet50 (He et al., 2016). These embeddings are clustered via K-means ($k = 10$) to group similar regions. A GMM with 5 components is fitted per cluster, and Fisher Vectors are computed as gradients of the log-likelihood.

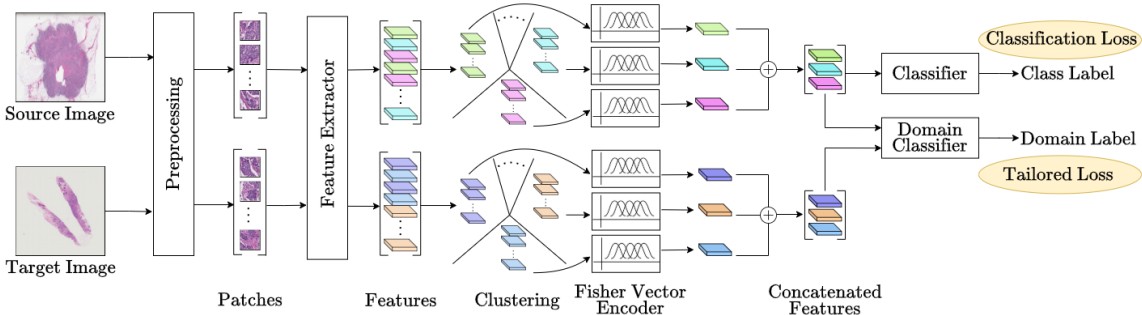

Figure 1: Patch embeddings are clustered and encoded using Fisher Vectors, then refined via adversarial domain adaptation

Fisher Vectors from all clusters are concatenated into a high-dimensional, permutation-invariant WSI representation, used for classification via MLP, AMIL (Ilse et al., 2018), ConvNeXt (Liu et al., 2022), or Swin Transformer (Liu et al., 2021), trained with cross-entropy loss. To handle domain shift, we apply adversarial adaptation following the DANN framework (Ganin and Lempitsky, 2015). A domain discriminator $D$ distinguishes source and target FVs, while the feature generator $G$ is trained to confuse $D$, optimizing the adversarial loss:

$$\mathcal{L}_{adv} = \mathbb{E}_{x_s \sim P_S}[\log D(G(x_s))] + \mathbb{E}_{x_t \sim P_T}[\log(1 - D(G(x_t)))] \tag{1}$$

Our total objective incorporates classification and domain alignment:

$$L = \min_N L_{cls}(x_{s_i}, y_{s_i}) - \lambda L_{adv}(x_s, x_t) + \gamma L_{MCC} + \eta L_{PLMMD} \tag{2}$$

where $L_{MCC}$ and $L_{PLMMD}$ enhance class separation and inter-domain distribution alignment.

The Pseudo-Label Maximum Mean Discrepancy (PLMMD) loss, which leverages pseudo-labels from target samples to strongly condition feature alignment on classes. PLMMD adapts the standard MMD loss by incorporating weights $w_{XX}$, $w_{XY}$, and $w_{YY}$ based on pseudo-labels:

$$L_{PLMMD} = w_{XX} \, \mathbb{E}_P[k(X, X)] - 2w_{XY} \, \mathbb{E}_{P,Q}[k(X, Y)] + w_{YY} \, \mathbb{E}_Q[k(Y, Y)]. \tag{3}$$

The weights are derived by normalizing source and target labels to mitigate class imbalances, computing dot products for instance relationships, and normalizing by the common class count. This training objective promotes domain-invariant embeddings and enhances target generalization.

## 3. Results and Discussion

We evaluate our framework on HER2 classification using TCGA-BRCA and Warwick datasets under two transfer settings. ResNet34 and MoCoV3 features are compared across three setups: no adaptation, adversarial adaptation, and our PLMMD+MCC-based tailored loss.

Table 1: HER2 classification results: TCGA-BRCA (source) to Warwick (target)

| Feature Extractor | Domain Adaptation | Tailored Loss | Accuracy |
|---|---|---|---|
| ResNet34 | No | No | 0.68 |
| ResNet34 | Yes | No | 0.76 |
| ResNet34 | Yes | Yes | **0.79** |
| MoCoV3 | No | No | 0.72 |
| MoCoV3 | Yes | No | 0.75 |
| MoCoV3 | Yes | Yes | **0.78** |

Table 1 shows that in the TCGA-BRCA → Warwick transfer, MoCoV3 outperforms ResNet34 without adaptation (0.72 vs. 0.68), indicating better baseline generalization. ResNet34 improves significantly with adaptation (0.76) and further with the tailored PLMMD+MCC loss (0.79), while MoCoV3 reaches 0.78. This suggests MoCoV3 learns more domain-invariant features initially, whereas ResNet34 benefits more from alignment. In the reverse

Table 2: HER2 classification results: Warwick (source) to TCGA-BRCA (target)

| Feature Extractor | Domain Adaptation | Tailored Loss | Accuracy |
|---|---|---|---|
| ResNet34 | No | No | 0.64 |
| ResNet34 | Yes | No | 0.70 |
| ResNet34 | Yes | Yes | **0.72** |
| MoCoV3 | No | No | 0.67 |
| MoCoV3 | Yes | No | 0.68 |
| MoCoV3 | Yes | Yes | **0.69** |

setting (Table 2), ResNet34 improves from 0.64 to 0.72 with adaptation and tailored loss, while MoCoV3 increases from 0.67 to 0.69. These results highlight the consistent advantage of our domain adaptation strategy, especially for supervised features.

Our approach combines self-supervised learning, Fisher Vector encoding, and tailored adaptation for robust cross-domain WSI classification. MoCoV3 generalizes well, while ResNet34 benefits more from PLMMD+MCC, reflecting a trade-off between robustness and adaptability. The modular design supports scalability, with future work targeting end-to-end training and multi-task extensions.

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
