# OpenReview forum: "Whole Slide Image Domain Adaptation Tailored with Fisher Vector, Self Supervised Learning, and Novel Loss Function"
_MIDL.io/2025/Short_Papers — MIDL 2025 - Short Papers_

### Official Review · Reviewer_t7p5 · 2025-04-24

**Rating:** 4
**Confidence:** 4

**Summary:**

This paper proposes a domain adaptation framework for robust WSI classification, effectively integrating self-supervised learning, clustering, and Fisher Vector encoding. The method demonstrates significant improvements over baseline approaches. Given these contributions, the work warrants acceptance for short paper presentation.

**Strengths:**

improvement over baselines
evaluation on public data

**Weaknesses:**

technical novelty is limited as it combines prior work.
not clear if the code will be available

---

### Decision · Program_Chairs · 2025-05-01

Accept